# StreamDiT: Real-Time Streaming Text-to-Video Generation

## Abstract

Recently, great progress has been achieved in text-to-video (T2V) generation by scaling transformer-based diffusion models to billions of parameters, which can generate high-quality videos. However, existing models typically produce only short clips offline, restricting their use cases in interactive and real-time applications. This paper addresses these challenges by proposing StreamDiT, a streaming video generation model. StreamDiT training is based on flow matching by adding a moving buffer. We design mixed training with different partitioning schemes of buffered frames to boost both content consistency and visual quality. StreamDiT modeling is based on adaLN DiT with varying time embedding and window attention. To practice the proposed method, we train a StreamDiT model with 4B parameters. In addition, we propose a multistep distillation method tailored for StreamDiT. Sampling distillation is performed in each segment of a chosen partitioning scheme. After distillation, the total number of function evaluations (NFEs) is reduced to the number of chunks in a buffer. Finally, our distilled model reaches real-time performance at 16 FPS on one GPU, which can generate video streams at 512p resolution. We have evaluated our method through both quantitative metrics and human evaluation. Our model enables real-time applications, *e.g.* streaming generation, interactive generation, and video-to-video.

## 1 Introduction

Video generation is a fundamental and trending problem in computer vision, with rapid progress in recent years. Diffusion models have been adopted for text-to-video (T2V) generation, and demonstrated capabilities of generating high-quality videos, with a potential as world simulators. The success was built upon the scalability of transformer-based architectures and huge compute resources to train billions of parameters. Despite achievements, many existing models are designed to generate short video clips, due to the high cost of scaling. As a result, generating long videos with low latency remains an extreme challenge Li et al. (2024), which is demanded in real-time and interactive applications.

Modern T2V models Polyak et al. (2024); Kong et al. (2025); Ma et al. (2025) are based on Diffusion Transformers (DiTs) Peebles and Xie (2023). They use auto-encoders to compress a video clip to a 3D latent. The auto-encoders are based on CNN layers with uniform compression over spatial and temporal domains. Therefore, the latent frames are separable. The models predict all frames together by gradually removing noise from a latent. Increasing video length is expensive due to the quadratic complexity of transformers. To make diffusion models capable of generating long videos, some approaches Xie et al. (2024); Yin et al. (2025) attempted to combine autoregression and diffusion, by either adding progressive noise to latent frames or causal masking in the final model. In an autoregressive manner, a latent frame is predicted from previous ones. We argue that recent high-quality video generation models benefit from full attention of all tokens in a sequence, which ensures high consistency in video content. Therefore, a method that preserves token communication is desired for enabling extendable video generation.

An inspiration comes from training-free methods for streaming generation, *e.g.* StreamDiffusion Kodaira et al. (2023) and FIFO-Diffusion Kim et al. (2024). StreamDiffusion utilizes pre-trained image diffusion models to edit an input video stream via batch denoising. It suffers from a lack of frame-to-frame consistency, leading to visual artifacts when generating continuous video streams.

FIFO-Diffusion enqueues a sequence of frames with different noise levels. It pops up a clear frame after one denoising step and enqueues a noise frame in the queue. Although they do not require training, the quality is limited. Another missing effort is sampling distillation, which is crucial for real-time applications. Since the queue is updated at every denoising step, popular distillation methods *e.g.* guided distillation Meng et al. (2023) and consistency distillation Luo et al. (2023) cannot be applied directly.

To address the challenges, we propose a complete solution for streaming video generation, named StreamDiT, which includes training and modeling. For training, we modify the flow matching Lipman et al. (2023) by introducing a moving buffer of frames. Within a buffer, we adopt full attention so that our method can be seamlessly applied to modern T2V models. At inference, the buffer is moving along the frame dimension, generating clean frames sequentially. Similar to previous work Xie et al. (2024); Kim et al. (2024), we allow buffered frames with different noise levels. We design a unified partitioning of latent frames with different chunk sizes and micro steps. We reveal that uniform noise used in original flow matching Lipman et al. (2023) and diagonal noise from previous work Kim et al. (2024); Xie et al. (2024) are special cases of our partitioning. With mixed training of different schemes, we can enhance video consistency and avoid overfitting on a specific scheme.

For modeling, we design our StreamDiT model architecture to fit its training process while being efficient for real-time applications. We modify the adaLN DiT Peebles and Xie (2023) by adding varying time embedding and replacing full attention with window attention Liu et al. (2021). We first train our StreamDiT as a basic T2V model, and then adapt it to streaming video generation. To achieve real-time applications, we also design a multistep distillation tailored for StreamDiT with a chosen partitioning scheme. With that, we distill our model to 8 sampling steps without classifier-free guidance (CFG). Finally, our model can generate video frames at 16 FPS on one H100 GPU, achieving real-time performance.

Our contributions are summarized as follows:

- We propose StreamDiT training for streaming video generation, which is based on flow matching by introducing a moving buffer. We design generalized partitioning of buffered frames that covers uniform noise, diagonal noise, and others in between. We also point out a requirement for generation models such that time embedding needs to be separable in the frame dimension.

- We design StreamDiT model as adaLN DiT with varying time embedding and window attention. Our StreamDiT has 4B parameters and is trained to generate videos at 512p resolution. We show that using mixed training with different partitioning schemes can improve the quality and flexibility of streaming video generation.

- We build a real-time solution using multistep distillation tailored for StreamDiT. Choosing a partitioning scheme that balances inference speed and quality, we distill micro steps to one for each segment. With all the efforts combined, our distilled model achieves real-time performance with 16 FPS on one GPU. In addition, we show some results to highlight potential real-time applications using our model.

## 2 RELATED WORK

### 2.1 TEXT-TO-VIDEO GENERATION

Arising with the development of text-to-image generation Dai et al. (2023), video generation bursts by introducing temporal modules. CogVideo Hong et al. (2023) adds temporal attention modules into a pre-trained autoregressive text-to-image model CogView2 Ding et al. (2022). Video Diffusion Models (VDM) Ho et al. (2022b) utilize a space-time factorized U-Net with joint image and video data training. Imagen Video Ho et al. (2022a) improves VDM with cascaded diffusion models for high-resolution video generation. Lumiere Bar-Tal et al. (2024) introduces a Space-Time U-Net architecture that generates the entire temporal duration of the video at once. Noticing the high computation of generation in pixel domain, researchers explored video generation in latent domain, *e.g.* Zhou et al. (2023); He et al. (2023); Blattmann et al. (2023); Guo et al. (2024); Chen et al. (2024a). Yet, these methods are based on UNet structures with temporal layers added to T2I models. As revealed by the work of DiT Peebles and Xie (2023), transformers show better scalability than UNets

in T2I generation. VDT Lu et al. (2024) proposes to use separated spatial and temporal attentions in a transformer block. GenTron Chen et al. (2024b) introduces a family of transformer-based diffusion models for image and video generation. Snap Video Menapace et al. (2024) proposes a transformer architecture with joint spatiotemporal blocks, which trained faster than U-Nets. ViD-GPT Gao et al. (2024) uses causal attention in temporal domain and frames as prompts, following the design of GPT in LLMs. Due to the limitations of compute, these models are trained at small or medium scales, up to a few of billion parameters, generating videos of a few seconds. In the latest work of MovieGen Polyak et al. (2024), the DiT model has been scaled to 30B parameters and can generate videos up 16 seconds, with high motion and aesthetic qualities.

Recently, the open-source community is also growing fast. OpenSora Zheng et al. (2024) initiated releases of small-sized T2V models with about 1B parameters. They use factorized transformers for spatial and temporal domains. Hunyuan Kong et al. (2025) video model has 13B parameters. It uses MMDiT Esser et al. (2024) with concatenated text and video tokens in self-attention. For auto-encoder, it adopts causal 3D convolution Yu et al. (2024) in its VAE. Step-Video-T2V Ma et al. (2025) with 30B parameters is by far the largest open-source model for video generation, which reduces the gap of model size between open-source and closed-source with similar quality.

## 2.2 Long Video Generation

The literature of long video generation can be categorized as training and training-free methods. Training new models or new modules injected to existing models requires a large amount of compute resources. NUWA-XL Yin et al. (2023) proposes a diffusion over diffusion architecture to generate long videos in a coarse-to-fine manner. StreamingT2V Henschel et al. (2024) proposes an autoregressive approach for long video generation, using a short-term memory block and long-term memory block of previous chunks. In Xie et al. (2024), a similar method was proposed by assigning latent frames with progressively increasing noise levels. VideoTetris Tian et al. (2024) trains a ControlNet branch of condition frames for long video generation. A recent work by Loong Wang et al. (2024) proposes an autoregressive LLM-based approach for video generation. The model is trained on 10-second videos and can be extended to one-minute long. Yet, the current quality of autoregressive models is not comparable with diffusion models.

For training-free methods, Gen-L-Video Wang et al. (2023) discovers that the denoising path of a long video can be approximated by joint denoising of overlapping short videos in the temporal domain. MimicMotion Zhang et al. (2024) extends MultiDiffusion Bar-Tal et al. (2023) to the temporal domain. It blends frames within overlap region progressively along the denoising process. Video-Infinity Tan et al. (2024) distributes a long-form video task to multiple GPUs, with dual-scope attention that modulates temporal self-attention to balance local and global contexts efficiently across the devices. FreeNoise Qiu et al. (2024) proposes a simple solution to generate consistent long videos by rescheduling a sequence of noises for long-range correlation and performing temporal attention over them by window-based function. Reuse-and-Diffuse Gu et al. (2023) proposes an iterative denoising for T2V, with staged guidance. It reuses intermediate denoising results from the previous clip to condition the current clip. FIFO-Diffusion Kim et al. (2024) proposes an inference technique that iteratively performs diagonal denoising, which concurrently processes a series of consecutive frames with increasing noise levels in a queue.

## 2.3 Sampling Efficiency

Distillation of sampling steps has been widely adopted for accelerating diffusion models. Compared to advanced numerical solvers, sampling distillation demonstrates better quality with fewer number of sampling steps. Progressive distillation Salimans and Ho (2022) distills two denoising steps of a teacher model to one step of a student model. Guided distillation Meng et al. (2023) proposed to first distill two function evaluations of CFG to one, and then perform progressive distillation on sampling steps. Consistency distillation Song et al. (2023); Luo et al. (2023) maps any points on a diffusion trajectory to the same origin, and therefore reduces the number of steps needed for sampling. Multistep consistency models Heek et al. (2024) splits the diffusion process to multiple segments, and consistency distillation can be applied at each segment. By allowing more budget on sampling steps, Multistep consistency distillation can achieve higher quality.

To further improve sampling efficiency, one-step approaches have been proposed, including UFO-Gen Xu et al. (2024b) and DMD Yin et al. (2024). The idea is to directly match distributions without

the Gaussian assumption. They require to hold multiple models (discriminator or score networks) during training, posing challenges for applying to large video models.

# 3 STREAMDIT TRAINING

## 3.1 BUFFERED FLOW MATCHING

**Flow Matching:** Our method is based on the Flow Matching (FM) Lipman et al. (2023) framework for training. FM produces a sample from the target data distribution by progressively transforming a sample from an initial prior distribution, such as a Gaussian. During training, for a data sample denoted as $\mathbf{X}_1$, we sample a time step $t \in [0, 1]$, and noise $\mathbf{X}_0 \sim \mathcal{N}(0, 1)$. These are then used to create a training sample $\mathbf{X}_t$. FM predicts the velocity $\mathbf{V}_t$ that moves the sample $\mathbf{X}_t$ in the direction of data sample $\mathbf{X}_1$.

A simple linear interpolation or the optimal transport (OT) path Lipman et al. (2023) is used to construct $\mathbf{x}_t$, *i.e.*,

$$\mathbf{X}_t = t\,\mathbf{X}_1 + (1 - (1 - \sigma_{\min})t)\,\mathbf{X}_0, \tag{1}$$

where $\sigma_{\min}$ is the standard deviation of $x$ at $t = 1$. Thus, the ground truth velocity can be derived as

$$\mathbf{V}_t = \frac{d\mathbf{X}_t}{dt} = \mathbf{X}_1 - (1 - \sigma_{\min})\mathbf{X}_0. \tag{2}$$

It is worth noting that this target is irrelevant to time step $t$. With parameters $\Theta$ and text prompt $\mathbf{P}$, the predicted velocity is written as $u(\mathbf{X}_t, \mathbf{P}, t; \Theta)$, and the training objective is represented as

$$\mathbb{E}_{t, \mathbf{X}_t} \| u(\mathbf{X}_t, \mathbf{P}, t; \Theta) - \mathbf{V}_t \|^2. \tag{3}$$

At inference, an FM model predicts the velocity to a clean sample on every denoising step. With the Euler solver, the inference can be formulated as

$$\mathbf{X}_{t+\Delta t} = \mathbf{X}_t + u(\mathbf{X}_t, \mathbf{P}, t; \Theta)\Delta t, \tag{4}$$

where $\Delta t$ is the step size.

**Buffered Flow Matching:** We consider streaming video generation as a sequence of (possibly latent) frames $[f_1, f_2, \ldots, f_N]$, and $N$ can be infinite. For a video diffusion model with frame buffer $B$, the clean data sample starting with frame $f_i$ is denoted as $\mathbf{X}_1^i = [f_i, \ldots, f_{i+B}]$. We allow different noise levels to the frames: $\tau = [\tau_1, \ldots, \tau_B]$ as a monotonically increasing sequence. Thus a training example can be constructed as

$$\mathbf{X}_\tau^i = \tau \circ \mathbf{X}_1^i + (1 - (1 - \sigma_{\min})\tau) \circ \mathbf{X}_0, \tag{5}$$

where $\circ$ denotes element-wise product. Please note that the noise sample $X_0$ remains the same.

At inference, the buffer is updated by model predicted flow

$$\mathbf{X}_{\tau+\Delta\tau}^i = \mathbf{X}_\tau^i + u(\mathbf{X}_\tau^i, \mathbf{P}, \tau; \Theta) \circ \Delta\tau, \tag{6}$$

where $\Delta\tau$ is a sequence of step sizes. If one or more frames achieve the final denoising step, they are popped out of the buffer, and new noise frames are pushed at the beginning of the buffer. Therefore, it can generate streaming video sequences.

## 3.2 PARTITIONING SCHEME

To facilitate flexible training and inference of StreamDiT, we design a unified partitioning of buffered frames. As illustrated in fig. 1, the buffer is partitioned to $K$ reference frames and $N$ chunks. Each chunk has $c$ frames and $s$ micro denoising steps.

**Clean Reference Frames:** To enhance temporal consistency, we can optionally cache the last $K$ fully denoised frames at the beginning of the buffer. We refer to these clean frames of as reference frames. They participate in the denoising step as input but are no longer denoised. The reference frames are updated in the same way as other frames when the buffer moves. Allowing optional reference frames matches the design of FIFO-Diffusion Kim et al. (2024), which can be viewed as a special

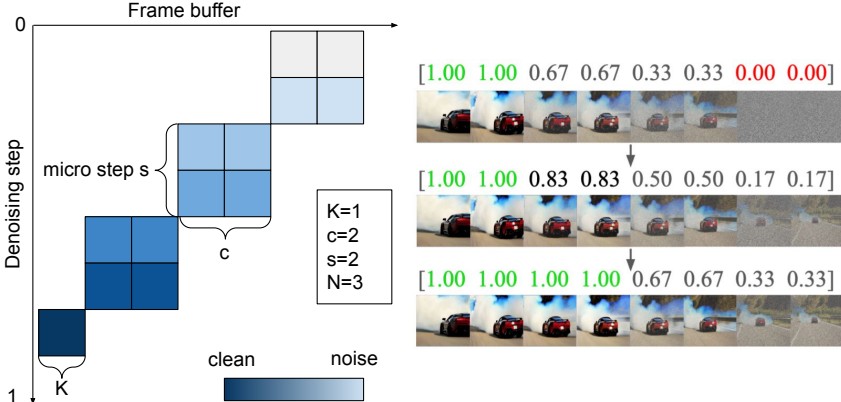

Figure 1: Illustration of StreamDiT partitioning. We partition the buffer to $K$ reference frames and $N$ chunks. Each chunk has $c$ frames and $s$ micro denoising steps.

case. Since our method is trainable, we found that reference frames can be skipped for streaming video generation. Hence, we set $K = 0$ in the rest of our work.

**Chunked Denoising:** Instead of denoising frame by frame, we group frames into stream chunks, with each chunk containing a specified number of frames indicated by chunk size. Noise levels are now applied at the chunk level, and each time a chunk of frames exits the pipeline. Let $N$ denote the number of stream chunks and $c$ the number of frames in each chunk. Then the total number of frames processed at any time is $K + N \times c$, and the number of denoising steps is constrained to $N$.

**Micro Step:** As the total number of denoising steps is bounded by $N$, however, for better performance, additional denoising steps are usually necessary. A straightforward approach to address this limitation is to increase the size of the buffer, but this is limited by the maximum capacity of the model to denoise frames and would also increase latency for each frame in the buffer.

To overcome this, we introduce an additional dimension of the design called micro-denoising step, illustrated in fig. 1. The core idea of micro step is to denoise stream chunks along the temporal axis while stagnating at a fixed spatial position for certain time. Let $s$ denote the denoising steps per micro step; then each stream chunk undergoes $s$ denoising steps before advancing to the next noise level and moving toward the output. This modification effectively extends the total number of denoising steps to $s \times N$ without increasing the buffer size.

With the incorporation of reference frames, chunked frames, and micro-step denoising, the following equations hold:

$$B = K + N \times c, \qquad T \quad = s \times N \tag{7}$$

where $B$ is the total length of the frame fed into the model, $N$ is the number of stream chunks, $c$ is the number of frames in each stream chunk, and $T$ represents the total number of inference steps.

**Mixed Training:** StreamDiT unifies original diffusion or FM with uniform noise and diagonal noise used by Kim et al. (2024); Xie et al. (2024). The latter enables streaming generation but hurts consistency of generated content. To enhance consistency and avoid overfitting, we adopt mixed training with different schemes. This drives the model to learn generalized denoising with different noise levels, instead of memorizing fixed noise levels. It is worth noting that our mixed training covers diffusion and FM training. Therefore, our model can work as a standard T2V generation without streaming. This is also used for initializing our streaming generation.

According to fig. 1, frames in each chunk correspond to a distinct segment of the overall time step range. Thus, during training, we sample a random time step for the $i$-th chunk as follows:

$$\tau_i \sim \text{Uniform}\left(\left[\frac{T}{N} \cdot (i-1), \frac{T}{N} \cdot i\right]\right) \tag{8}$$

Interestingly, the StreamDiT training can be viewed as parallel training of the full range of denoising.

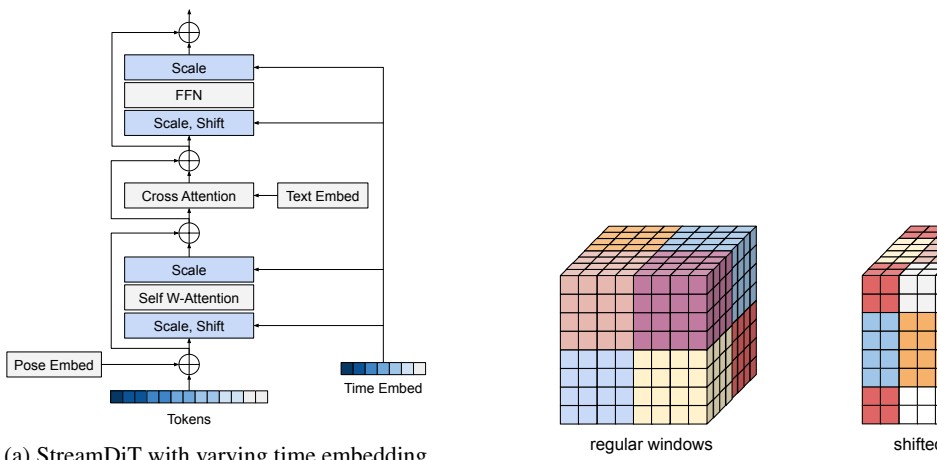

(a) StreamDiT with varying time embedding and window attention.

(b) Window partitioning: regular vs. shifted.

regular windows        shifted windows

Figure 2: (a) Modified adaLN DiT with varying time embeddings; (b) window partitioning variants.

## 4 STREAMDIT MODELING

### 4.1 MODEL ARCHITECTURE

**Time-Varying DiT:** As shown in section 3.1, our StreamDiT changes the scalar condition $t$ to a sequence $\tau$ in the model. This requires that the time condition should be separable in the frame dimension. We follow the standard adaLN DiT Peebles and Xie (2023) architecture, where time embedding is used for scale and shift modulations, and modify it with varying time embedding. Specifically, we reshape the latent tensor to 3D: $[F, H, W]$, and apply time embedding along the first dimension. As shown in fig. 2a, we modify the adaLN DiT with varying time embedding. Input tokens are also corrupted with different levels of noise.

**Window Attention:** To make the DiT model more efficient, we adopt window attention Liu et al. (2021) for all self-attention layers. Specifically, we partition a 3D latent with shape $[F, H, W]$ to non-overlapped windows with size $[F_w, H_w, W_w]$, and apply masking when computing self-attention, such that a token can only see tokens within the same window. To ensure consistency across windows, we shift by half of the window size every other layer. As illustrated in fig. 2b, voxels at axis ends will be warped over to the beginnings in the shifting. Accumulatively, global token communication can broadcast to all tokens, while each computation is efficient with local attention. The complexity of window attention is only $\frac{F_w \times H_w \times W_w}{F \times H \times W}$ of full attention.

**Other Components:** Since we are targeting real-time applications, we chose a moderate size (4B) for our DiT. We reuse a temporal auto-encoder (TAE) in Movie Gen Polyak et al. (2024) with compression rate 4 in temporal domain and 8 in spatial domain. The latent channel size is 8, with the consideration that a moderate latent space is easier to learn for small generation models. The trade-off is that with a smaller temporal compression rate, the model needs to generate more latent frames to reach the same FPS.

For text encoders, we adopt the same ones from Movie Gen Polyak et al. (2024), including UL2 Tay et al. (2023), ByT5 Xue et al. (2022), and MetaCLIP Xu et al. (2024a). The text encoders run fast on GPU. We only run text encoders when the prompt is changed for streaming video generation. Therefore, its computation time is negligible.

### 4.2 MULTISTEP DISTILLATION

Sampling distillation is a key component to build real-time streaming video generation. Due to the modifications of frame partitioning and micro denosing steps made by StreamDiT, standard sampling distillation methods Salimans and Ho (2022); Meng et al. (2023); Heek et al. (2024) cannot be applied. Thus, a customized sampling distillation for StreamDiT is needed.

As described in section 3.2, StreamDiT partitions frames in a buffer to $N$ chunks, each of which has $s$ denoising steps before moving the buffer. In particular, we aim at reducing the micro-step $s$

| | Subject Consistency | Background Consistency | Temporal Flickering | Motion Smoothness | Dynamic Degree | Aesthetic Quality | Quality Score |
|---|---|---|---|---|---|---|---|
| ReuseDiffuse | 0.9501 | 0.9615 | 0.9838 | 0.9912 | 0.2900 | 0.5993 | 0.8019 |
| FIFO | 0.9412 | 0.9576 | 0.9796 | 0.9889 | 0.3094 | 0.6088 | 0.7981 |
| Ours | 0.9622 | 0.9625 | 0.9671 | 0.9861 | 0.5240 | 0.6026 | **0.8185** |
| Ours-distill | 0.9491 | 0.9555 | 0.9649 | 0.9831 | 0.7040 | 0.5940 | 0.8163 |

Table 1: VBench quality metrics of our evaluation. Our models outperform others, and our distilled model is close to our teacher model.

in eq. (7). Our StreamDiT model is trained with mixed partitioning schemes, to make it flexible to different scenarios. In distillation, we first choose a partitioning scheme. Specifically, we set $K = 0$, $c = 2$, $s = 16$, $N = 8$ for the teacher model. This indicates that the teacher model has $s * N = 128$ denoising steps with CFG. The FM trajectory is split to $N$ segments. We then perform step distillation in each segment separately. In practice, we perform both step and guidance distillation at the same time, by distilling multiple CFG steps of the teacher into a single conditional forward pass.

For the multistep distillation, we reduce the micro step $s$ to 1, resulting in $N$-step sampling without CFG, which still follows the design of StreamDiT for streaming. Moreover, it powers up a significant speed-up that enables real-time video generation.

## 5 EXPERIMENTS

### 5.1 IMPLEMENTATION DETAILS

We finetune a T2V model Polyak et al. (2024) with 4B parameters using the architecture introduced in section 4.1. The latent size is $[16, 64, 64]$. With a $[4\times, 8\times, 8\times]$ TAE, the base model generates 64 frames at 512p resolution.

Then we adapt the model for streaming video generation. Our StreamDiT training has three stages: task learning, task generalization, and quality fine-tuning. In the first stage, we use a small amount of high-quality video data (3K videos) with a large learning rate (1e-4) to adapt the original T2V into a video streaming model. The second stage involves further training on the pretraining dataset (2.6M videos) with a small learning rate (1e-5) to improve generalization for video streaming. In the final stage, we finetune the model on the high-quality video dataset with a small learning rate (1e-5) to optimize output quality. Each stage is trained with 10K iterations on 128 NVIDIA H100 GPUs.

To achieve real-time inference, we additionally apply the multistep distillation described in section 4.2. We found a partitioning scheme $c = 2$, $s = 16$, $N = 8$ is good for our multistep distillation considering both quality and efficiency, so we used it for teacher model inference. The distillation is also conducted on the 3K high-quality video dataset and 64 NVIDIA H100 GPUs for 10K iterations. After distillation, the micro step is reduced to 1, and the total number of sampling steps is 8.

### 5.2 EVALUATION

We compare our method with ReuseDiffuse Gu et al. (2023) and FIFO-Diffusion Kim et al. (2024) for streaming generation of long videos. To make a fair comparison, we implemented their methods on our base model, so they share the same visual quality. We select 50 prompts from the evaluation dataset of Movie Gen Polyak et al. (2024) that are suitable for long videos. We adopt VBench Huang et al. (2024) quality metrics for quantitative evaluation.

As shown in table 1, our models outperform others with high quality scores. All methods have similar aesthetic quality and imaging quality scores. This is because they are derived from the same base model. ReuseDiffuse and FIFO achieve higher temporal consistency and motion smoothness. However, by examining their generated videos, the content is more static. This is also reflected in the dynamic degree column, where our models are much better.

In addition to VBench metrics, we also conduct human evaluations following the guidance in Polyak et al. (2024). A human evaluation compares two model results on the same prompts side-by-side. Annotators were asked to evaluate pairs of video samples to determine which one was superior or if they were of comparable quality along several axes. The evaluation criteria encompassed four aspects: overall quality, frame consistency, motion completeness, and motion naturalness. For these

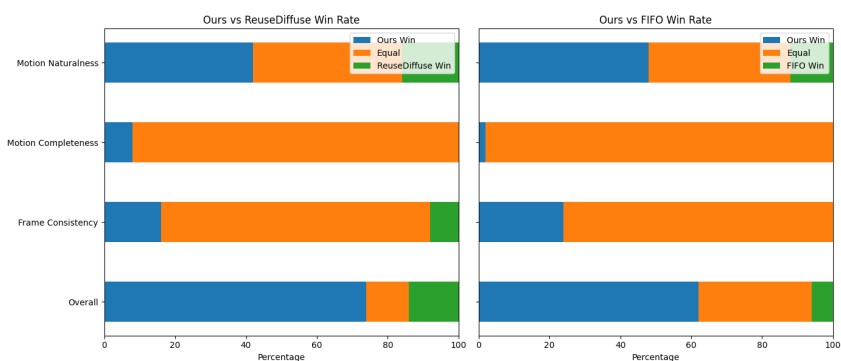

Figure 3: Human evaluations of our method compared with others, where our model shows higher win rates across all axes.

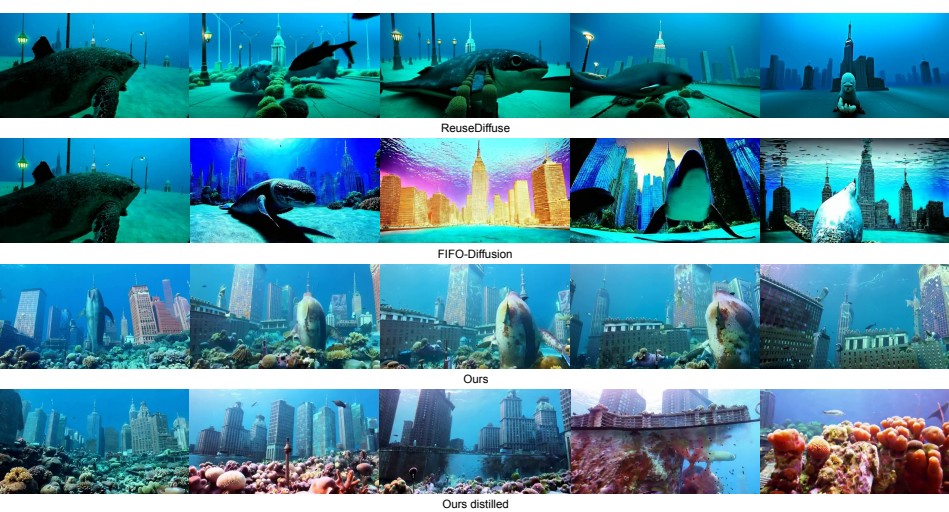

Figure 4: Visual results selected from our evaluation. Our models show better consistency and higher quality than others. Our distilled model has a similar quality with our teacher model.

evaluations, we employ the same set of 50 prompts used in the VBench evaluation, generating 8-second videos at a resolution of 512p for each prompt. As shown in fig. 3, our proposed method surpasses existing approaches across all evaluation metrics.

We show some selected frames from generated videos in fig. 4. The videos are one-minute long. We can see that our models have more consistent content with more motions, while others are static. This observation aligns with the VBench metrics.

## 5.3 Ablation Study

We conduct an ablation study to analyze the mixed training discussed in section 3.2. Specifically, we train models with different mixing schemes as shown in table 2. Chunk size 1 represents the Progressive AR Diffusion Xie et al. (2024) implemented on our model. Recall that our base model has 16 latent frames, so chunk size 16 is the basic T2V. After the models are trained, we generated

| Chunk size | [1] | [1,2] | [1,2,4] | [1,2,4,8] | [1,2,4,8,16] |
|---|---|---|---|---|---|
| Quality score | 0.8129 | 0.8100 | 0.8080 | 0.8076 | **0.8144** |

Table 2: Ablation of mixed training. Chunk size 1 represents the Progressive AR Diffusion Xie et al. (2024), and chunk size 16 represents the original T2V without streaming. A mixed training with all chunk sizes show the best quality score.

videos using chunk size $c = 1$, since this is the case covered by all models. Among them, we see that the mixture of all chunk sizes achieves the best quality score, although the inference is biased to chunk size 1 as being set in inference. Please note that we use a different inference scheme in table 1, so the numbers of our model are different. The first one with chunk size 1 and no mixing achieves the second-best quality score, indicating an overfitting to this case.

### 5.4 APPLICATIONS

**Real-Time Streaming:** Our distilled model has a chunk size 2 and 8 sampling steps without CFG. We benchmark its performance on one H100 GPU. It takes 482 ms for one denoising step to generate 2 latent frames and 8 video frames after a $4\times$ TAE. The latency of text encodering and TAE decoding are negligible. Thus, our distilled model can reach real-time performance at **16 FPS**.

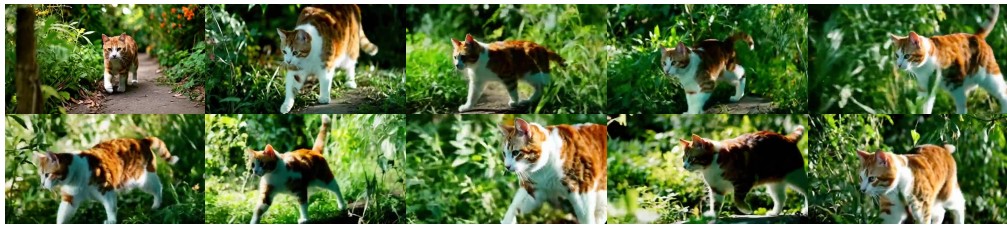
White and orange tabby cat runs through a dense garden.

Figure 5: We generate a 5-minute video to demonstrate the potential for infinite streaming.

**Infinite Streaming:** To explore the ability of infinite streaming, we try our distilled model to generate a video over 5 minutes long. As shown in fig. 5, the video content and quality are consistent after a very long generation, demonstrating the potential for infinite streaming.

**Interactive Streaming:** Additionally, we showcase the interactive storytelling capabilities of our model using a sequence of semantically related prompts. The qualitative results in fig. 6 demonstrate that our model effectively controls video events interactively, based on user-specified prompts.

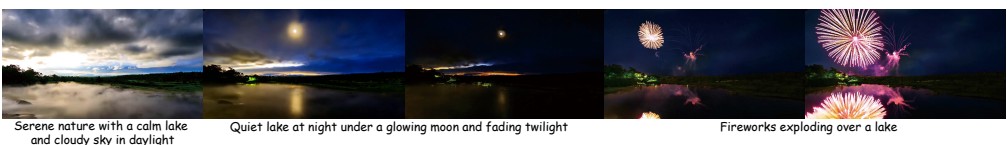
Serene nature with a calm lake and cloudy sky in daylight    Quiet lake at night under a glowing moon and fading twilight    Fireworks exploding over a lake

Figure 6: Interactive video streaming. The user can enter prompts to navigate the generated video.

## 6 CONCLUSION AND LIMITATIONS

In this work, we present StreamDiT for streaming video generation. It includes a novel training framework and efficient modeling that can run in real-time. StreamDiT training is based on flow matching with a moving buffer. We design generalized partitioning that unifies different schemes. Through experiments, we show that mixed training with different schemes can improve the consistency and quality of generations. To achieve real-time performance, we train an efficient StreamDiT model based on time-varying DiT with window attention. We further distill the model to 8 sampling steps, using the proposed multistep distillation tailored for StreamDiT. Our model achieves high temporal consistency across frames, addressing critical challenges in long-form and interactive video applications.

Our StreamDiT model has only 4B parameters, which is moderate compared to some closed-source and open-source foundation models. The model capacity therefore limits the quality of basic T2V generation. As a result, we noticed artifacts in some of the generated videos, *e.g.* the Janus problem of the cat in our 5-minute video, and motion artifacts of the rally car. When applying our method on a larger model (30B section C in Appendix) with a light training process, we found the quality improved a lot. From there, we can enhance the quality of small models via distillation for better real-time applications.

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

# Appendix

## A INFERENCE DETAILS

StreamDiT is specifically designed to achieve real-time responsiveness and interactivity, and its inference pipeline is structured accordingly. An overview of this pipeline is provided in fig. 7. In the main thread (Thread 1), the system performs the denoising operation, refills the stream queue, and emits denoised video frames from the queue to forward them to a separate decoder thread (Thread 2). This decoder thread runs concurrently, decoding the latent video frames to actual video frames. These resulting frames are then rendered in real time, allowing users to observe the changes immediately.

Additionally, a prompt callback function operates continuously on another thread (Thread 3), listening for new user prompts in real time. When a user provides a new prompt, it is converted into text embedding by text encoders, and the embedding is sent to the DiT thread to update the existing embedding. Subsequent denoising steps then use this updated embedding through a cross-attention mechanism, changing the direction of text guidance dynamically (fig. 9). This design enables users to interactively influence and modify video content in real time through prompt inputs.

In StreamDiT, since information from both preceding and succeeding video segments is always present in the context, it is essential to consider not only the explicit text guidance but also the implicit influence of video guidance. When the chunk size is small, the noise level difference between adjacent blocks in the Stream Queue decreases, amplifying the effect of video guidance. Therefore, to allow for larger content transformations, such as morphing, a larger chunk size is preferable. Conversely, if the goal is to maintain semantically continuous changes, such as variations in the walking direction of a character, a smaller chunk size is more suitable.

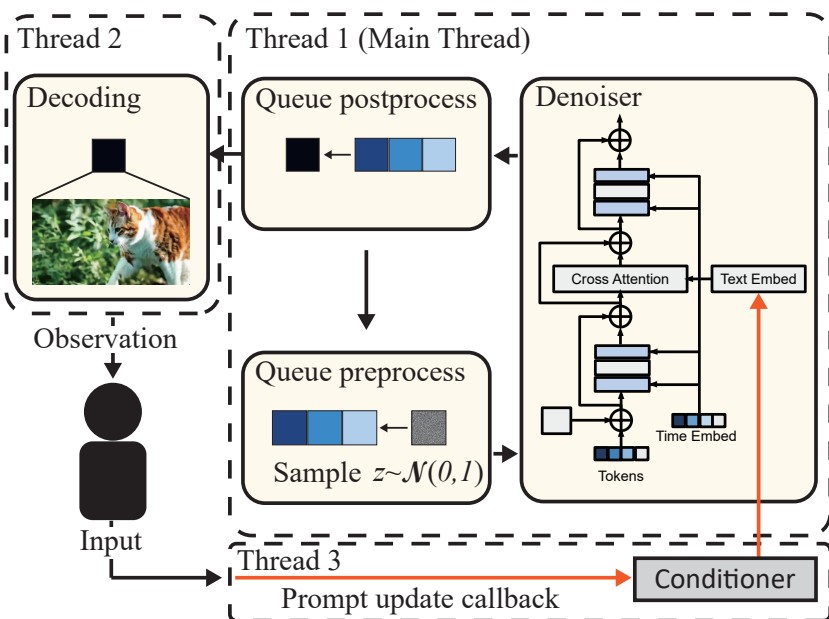

Figure 7: Interactive inference pipeline of StreamDiT: To decrease latency, generative models, decoder and text encoder are in separate process.

To perform stream denoising, the Stream Queue is filled with latent video frames that vary gradually in noise level. In the case where an initial video is provided in advance, such as in video-to-video scenarios, the video can first be encoded and then filled into the Stream Queue by adding Gaussian noise with appropriate stepwise noise levels. However, in the case of text-to-video generation, as shown in fig. 10, the process starts with chunk generation using a standard T2V model. During this stage, the intermediate video latents at each denoising step must be cached. Afterward, from the cached intermediate latents, those corresponding to the appropriate noise levels and video frames

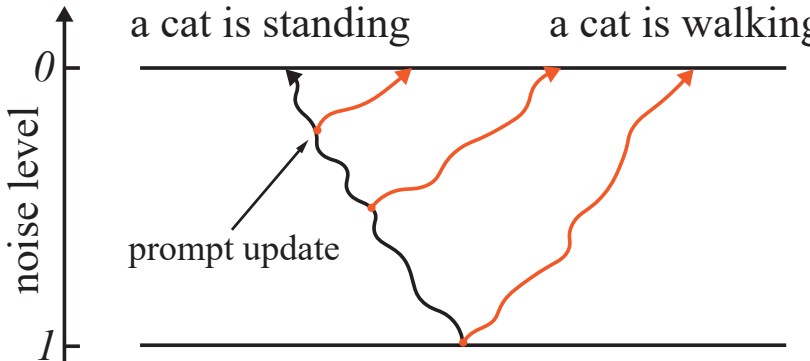

Figure 8: Real-time video streaming. With real-time streaming generation, our model can potentially work as a game engine.

Figure 9: Denoising Trajectory Change with Text Guidance Update: As denoising progresses toward the final stages, it becomes increasingly difficult to deviate from the outcome dictated by the original text guidance.

are selected in accordance with the StreamDiT inference configuration, and used to populate the Stream Queue. Once the queue is prepared, infinite-length video generation becomes possible by autoregressively and continuously performing stream denoising. Furthermore, as illustrated in the bottom row of fig. 10, when the number of stream chunks $N$ is set to one and one or more reference frames are used ($K \geq 1$), conventional chunk-based video extension is also possible. Because of StreamDiT's highly flexible unified architecture, it is possible to choose the chunk size and block size mixed during training according to the intended use. This enables a single model to flexibly switch between various inference patterns during deployment.

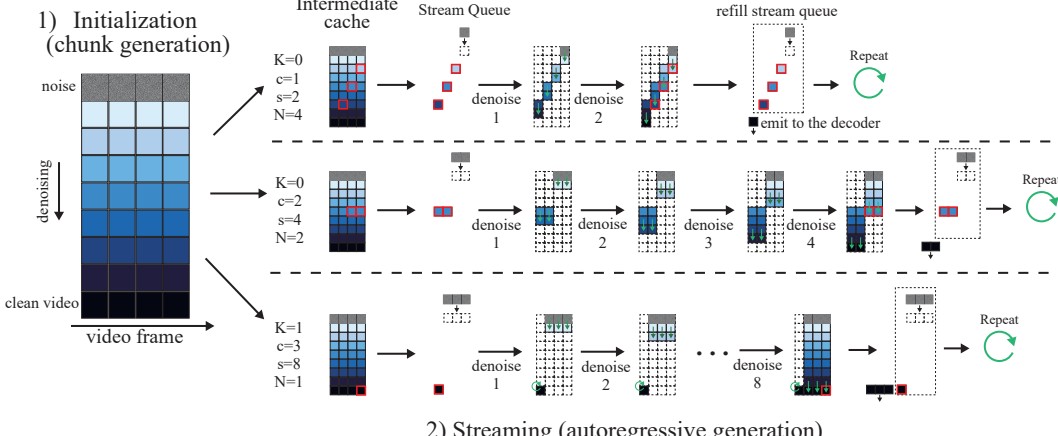

Figure 10: In the T2V scenario, StreamDiT first performs standard chunk generation to prepare intermediate latent cache. Once the intermediate cache is fully populated, the appropriate blocks are retrieved to construct the initial Stream Queue. Different inference configurations can be activated, provided that those configurations were included during mixed chunk training.

| Method | Scheme | Consistency | Streaming |
|--------|--------|-------------|-----------|
| Uniform | $c = B, s = 1$ | High | No |
| Diagonal | $c = 1, s = 1$ | Low | Yes |
| StreamDiT | $c = [1, \ldots, B], s = \frac{T}{N}$ | High | Yes |

Table 3: StreamDiT unifies different partitioning schemes.

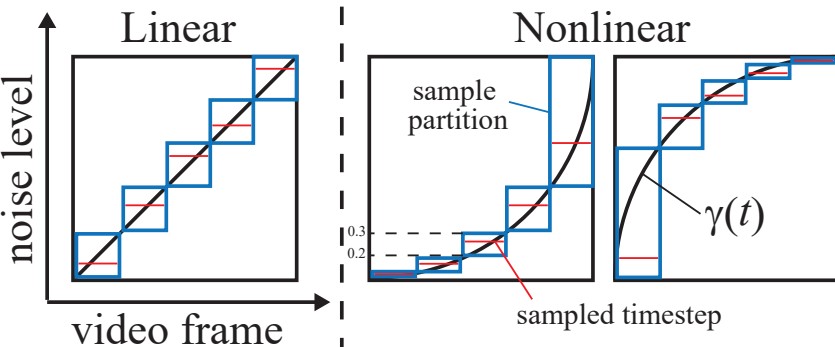

Figure 11: Partitioned Noise Training: The partitioning strategy can be defined by an arbitrary function—linear or nonlinear—that aligns with the chosen noise scheduling strategy.

## B    DESIGN CHOICES AND TRAINING DETAILS

Our design process begins by defining the global context length $B$ of the base text-to-video (T2V) model. Once $B$ is established, we determine a corresponding micro-step budget, enabling the informed selection of an appropriate chunk size $c$ to meet latency constraints. The choice of chunk size critically balances responsiveness and visual fidelity; larger chunk sizes yield more stable and higher-quality outputs but incur increased latency. To ensure flexibility across various inference scenarios, we adopt mixed-chunk training, which minimally impacts overall performance. Notably, incorporating the full-context chunk size ($c = B$) into mixed-chunk training substantially improves output quality by preserving key characteristics of the original T2V model, as demonstrated in table 2. For our distilled real-time model, we select a chunk size of $c = 2$ to achieve optimal performance at 16 FPS. Choosing a smaller chunk size ($c = 1$) would reduce performance to approximately 8 FPS in a step-distilled scenario. The micro-step size $s$ is not tuned independently; rather, it is directly determined by the chosen chunk size and the total number of denoising steps ($T = s \times N$).

In this paper, we trained StreamDiT using a partitioning strategy based on a *linear noise schedule*. However, depending on the denoising scenario, it may be beneficial to adopt a *nonlinear noise schedule* during inference. For instance, stronger denoising may be preferred in earlier frames, while detailed reconstruction might be prioritized in later ones. To enhance the performance of stream denoising under such conditions, it is crucial that the noise distribution during training closely matches the intended inference-time noise schedule (fig. 11). To achieve this, we define a general noise scheduling function $\gamma(t)$, mapping the normalized time domain $[0, 1]$ to the noise level range $[0, 1]$. Here, $t$ represents normalized temporal positions, such as frame indices or timestamps. We partition the time domain into $N$ equal segments, and within each segment $\left(\frac{i-1}{N}, \frac{i}{N}\right]$, we randomly select a time point $t_i$ from a uniform distribution. The noise level at this sampled point, $\gamma(t_i)$, is then applied uniformly across the entire interval. The resulting stepwise noise function $\hat{\gamma}(t)$ is expressed as:

$$\hat{\gamma}(t) = \sum_{i=1}^{N} \gamma(t_i) \cdot \mathbf{1}\left[t \in \left(\frac{i-1}{N}, \frac{i}{N}\right]\right],$$

$$\text{where } t_i \sim \text{Uniform}\left(\left[\frac{i-1}{N}, \frac{i}{N}\right]\right). \tag{9}$$

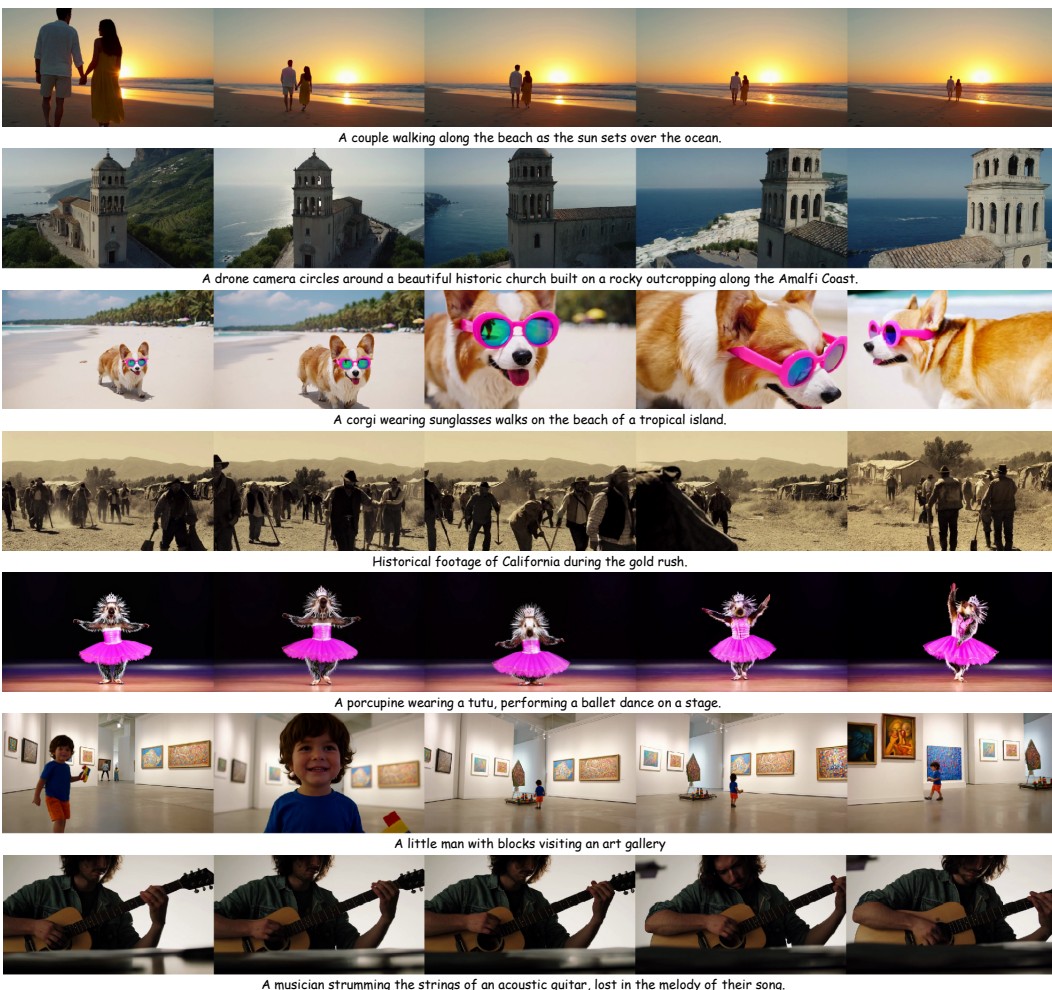

Figure 12: High-quality videos generated by the 30B model, demonstrating the scalability of our method.

This approach enables us to approximate various nonlinear noise schedules with an easily implementable step function. For example, a linear schedule corresponds to $\gamma(t) = t$, while exponential schedules can be modeled as $\gamma(t) = t^k$ with $k > 1$. Thus, our formulation provides flexibility to match different inference-time behaviors.

Furthermore, StreamDiT involves latent representations with different noise levels interacting within a shared context. Due to this design, the model is sensitive to the chosen noise scheduling strategy and the context length. Therefore, careful consideration is required when setting these parameters. For instance, if most frames are assigned high noise levels, these frames will convey significantly less information compared to frames with lower noise. This scenario effectively reduces the usable context size, restricts information flow across frames, and can increase the difficulty of training. Consequently, improper design of noise schedules can negatively impact the denoising quality and temporal consistency of the model's outputs. Hence, selecting an appropriate noise scheduling strategy is crucial for achieving optimal model performance.

## C  MORE RESULTS

To evaluate the capabilities in generating higher-quality videos, we further fine-tuned the 30B model from Movie Gen Polyak et al. (2024). While the 30B model is not available for real-time generation at the present, it effectively supports the creation of exceptionally high-quality videos, as illustrated

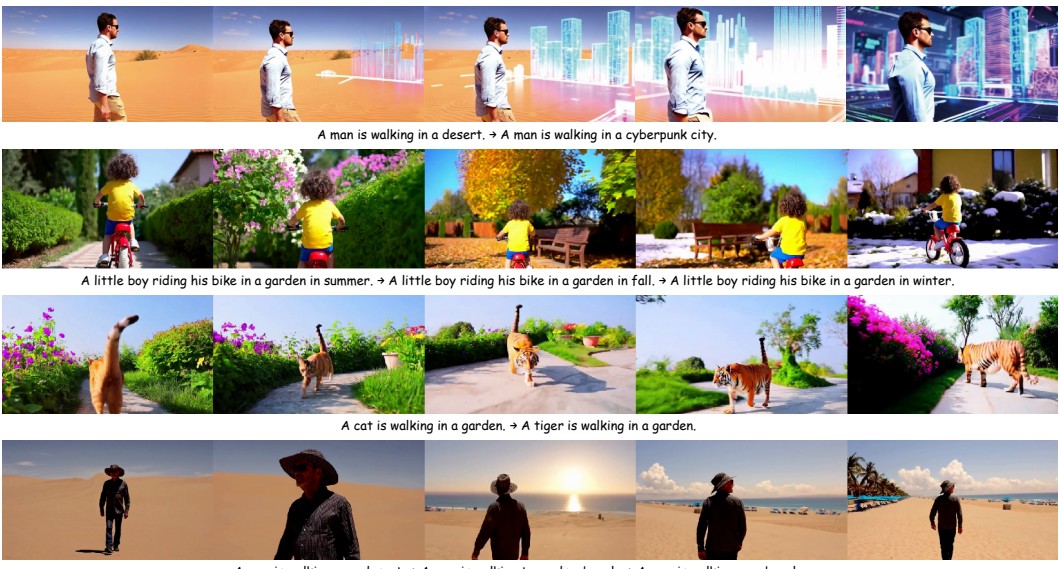

Figure 13: Sequential storytelling prompts can mitigate repetitive content and enable dynamic contents change.

| Config | | Quality | Subject | Background | Flickering | Motion | Dynamic | Aesthetic | Imaging |
|---|---|---|---|---|---|---|---|---|---|
| Reference frames | fixed 1 | 0.8175 | 0.9668 | 0.9769 | 0.9813 | 0.9910 | 0.4767 | 0.5500 | 0.6683 |
| | fixed 2 | 0.8202 | 0.9643 | 0.9763 | 0.9810 | 0.9911 | 0.5533 | 0.5482 | 0.6536 |
| | fixed 4 | 0.8151 | 0.9634 | 0.9748 | 0.9736 | 0.9886 | 0.5523 | 0.5292 | 0.6713 |
| | fixed 8 | 0.8222 | 0.9664 | 0.9768 | 0.9809 | 0.9913 | 0.5320 | 0.5538 | 0.6680 |
| | mixed 1 | 0.8228 | 0.9652 | 0.9765 | 0.9797 | 0.9907 | 0.5733 | 0.5421 | 0.6704 |
| | mixed 2 | 0.8244 | 0.9657 | 0.9766 | 0.9787 | 0.9903 | 0.6067 | 0.5410 | 0.6685 |
| | mixed 4 | 0.8230 | 0.9647 | 0.9769 | 0.9787 | 0.9902 | 0.5880 | 0.5413 | 0.6691 |
| | mixed 8 | 0.8246 | 0.9654 | 0.9761 | 0.9784 | 0.9901 | 0.6120 | 0.5418 | 0.6688 |
| T2V mix ratio | 0.1 | 0.8187 | 0.9690 | 0.9796 | 0.9823 | 0.9909 | 0.4852 | 0.5437 | 0.6693 |
| | 0.3 | 0.8218 | 0.9611 | 0.9761 | 0.9817 | 0.9913 | 0.5600 | 0.5497 | 0.6605 |
| | 0.5 | 0.8186 | 0.9626 | 0.9752 | 0.9770 | 0.9893 | 0.5778 | 0.5396 | 0.6600 |

Table 4: Full VBench scores of our ablation studies on the 30B model.

in fig. 12. Our model consistently produces extended videos characterized by impressive visual quality, content coherence, and diverse scenes aligned accurately with provided text prompts.

Additionally, ablation studies on the 30B model using VBench are summarized in table 4. The observed quality scores exhibit minimal variation, underscoring the stability and robustness of our method across various configurations.

Due to the inherent context-length limitations of base T2V models, even when augmented by autoregressive denoising improvements such as StreamDiT, the effective temporal context remains restricted. Consequently, repeatedly using the same prompt results in increasingly repetitive content, limiting the diversity of the generated videos. To address this limitation, we propose using a sequence of different story telling prompts during inference. This strategy maintains temporal coherence while allowing dynamic variation in the generated content (section A). As shown in fig. 13, sequential storytelling prompts significantly mitigate repetitive visual patterns and enhancing the feasibility in long-form video generation tasks. This approach facilitates the production of coherent videos with creative content, smooth object transformations, and seamless scene transitions.

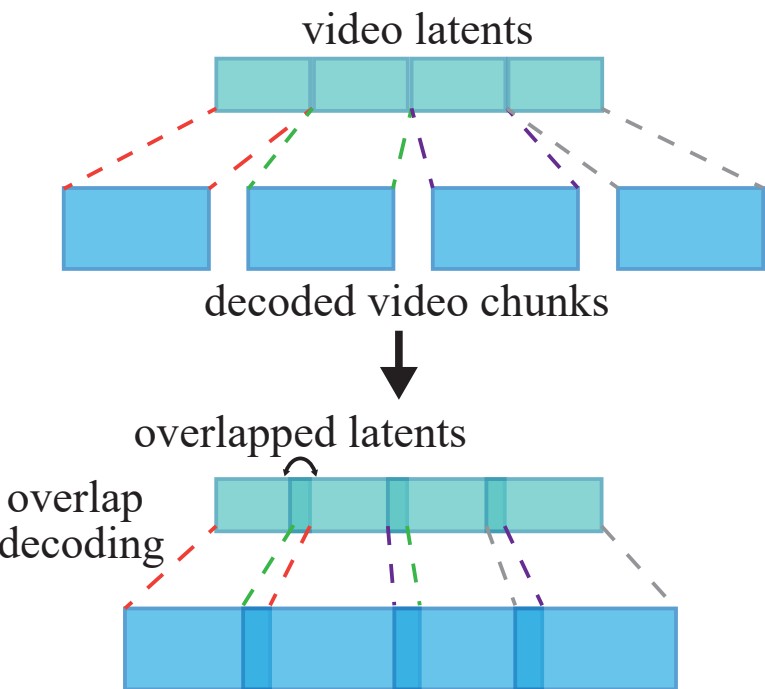

Figure 14: Overlap decoding

## D  LIMITATIONS

As described in section A, the effective context length of StreamDiT fundamentally depends on the base T2V model, and thus lacks long-term memory. When content falls outside the short-term memory window (*i.e.* context length) of StreamDiT, the associated information is likely to be lost. This can lead to issues such as identity mismatches in a person's face, or background inconsistencies when the camera makes a full rotation. However, since StreamDiT is orthogonal to additional long-term memory mechanisms, it is possible to address this issue in the future by combining it with long-term memory architectures such as State Space Models like Mamba Gu and Dao (2023).

Furthermore, with the current decoding strategy of StreamDiT, although the video frames are smoothly connected at the latent level, because video latent chunks are sequentially emitted from the Stream Queue and decoded separately, slight seams or flickering may be observed between decoded chunks in the final video. This occurs because a smooth connection at the latent level does not guarantee a seamless reconstruction in the decoded video. As a potential solution shown in fig. 14, when decoding, one could concatenate the end of a cached previous video latent to the beginning of a newly emitted video latent from the Stream Queue to create an extended chunk, which is then decoded. This overlapping strategy helps to reduce the appearance of seams.

The improvement of StreamDiT with those additional approaches is left as future work.

