# OpenReview forum: "StreamDiT: Real-Time Streaming Text-to-Video Generation"
_ICLR.cc/2026/Conference — ICLR 2026 Conference Withdrawn Submission_

### Official Review · Reviewer_s2Y7 · 2025-10-30

**Soundness:** 3
**Presentation:** 2
**Contribution:** 3
**Rating:** 4
**Confidence:** 4

**Summary:**

This paper proposes StreamDiT, a real-time text-to-video generation method that supports streaming and long video generation. The main contribution is to train a moving buffer-based denoising framework and distill the trained model to boost efficiency. Extensive experiments demonstrate the real-time performance of StreamDiT.

**Strengths:**

1. The performance of StreamDiT is promising and convincing. The generated visual results in the one-minute and five-minute long videos are visually appealing and demonstrate strong temporal consistency.

2. The mixed training scheme using a moving buffer and generalized partitioning is well-motivated and clear. Additional engineering optimizations such as the use of window attention for efficiency, are also valuable contributions.

**Weaknesses:**

1. The novelty appears limited. Can the authors clarify the fundamental difference between StreamDiT and teacher forcing/self-forcing methods? Both approaches involve manipulating noise schedules during training and share similar FIFO properties at inference. The innovation in this paper seems focused on the partitioning design and engineering optimizations. The dynamic timestep and inference strategy also appear to be a detailed engineering refinement of prior work like FIFO-Diffusion.

2. Comparisons are mainly made against training-free methods. As StreamDiT requires significant resources for a multi-stage training and distillation process, it seems unfair to compare its performance only against training-free methods like FIFO-Diffusion and the older baseline ReuseDiffuse. The paper should compare StreamDiT with other training-based streaming or long-video methods, such as Self-Forcing[1] or History-Guided Video diffusion [2] to better contextualize its contributions.


[1] Huang, Xun, et al. "Self Forcing: Bridging the Train-Test Gap in Autoregressive Video Diffusion." arXiv preprint arXiv:2506.08009 (2025).

[2] Song, Kiwhan, et al. "History-guided video diffusion." arXiv preprint arXiv:2502.06764 (2025).

**Questions:**

1. The paper demonstrates a 5-minute video generation. What is the practical limit for generating a temporally coherent video, and have the authors tested for longer durations (e.g., 10+ minutes)?

2. The paper states that the model "lacks long-term memory". How does the method handle or mitigate the loss of early context in very long videos, such as maintaining object identity or background consistency over extended periods?

---

### Official Review · Reviewer_sacD · 2025-10-30

**Soundness:** 3
**Presentation:** 3
**Contribution:** 3
**Rating:** 6
**Confidence:** 3

**Summary:**

This paper introduces StreamDiT, a 4-billion parameter streaming video generation model that addresses the limitations of existing text-to-video systems which only produce short clips offline. StreamDiT uses flow matching with a moving buffer, mixed training with different frame partitioning schemes, and adaptive layer normalization DiT architecture with varying time embeddings and window attention to achieve real-time video generation. Through a novel multistep distillation method that reduces function evaluations to match the number of buffer chunks, the model achieves 16 FPS performance on a single GPU at 512p resolution, enabling interactive applications like streaming generation, real-time interaction, and video-to-video transformation while maintaining content consistency and visual quality.

**Strengths:**

1. This paper enables real-time streaming video generation at 16 FPS on a single GPU, overcoming the offline-only limitation of existing text-to-video models.

2. It employs mixed training with different frame partitioning schemes to ensure both content consistency and high visual quality in generated video streams.

3. It introduces a tailored multistep distillation method that significantly reduces computational cost, making interactive applications practically feasible.

**Weaknesses:**

1. The paper only compares with U-Net-based methods such as ReuseDiffuse and FIFO. Please provide qualitative and quantitative comparison results with more advanced DiT-based methods.
2. When generating longer videos, such as the car video in "Real-Time Streaming Video Generation" from the supplementary materials, the video quality noticeably deteriorates as time progresses.
3. The examples of "Interactive Video Generation" provided in the paper all involve scene or appearance changes. How does StreamDiT perform when it comes to motion changes? Please provide relevant video results.

**Questions:**

Please refer to weaknesses.

---

### Official Review · Reviewer_E6RM · 2025-11-01

**Soundness:** 2
**Presentation:** 4
**Contribution:** 2
**Rating:** 2
**Confidence:** 5

**Summary:**

The paper proposes a natively trained streaming video generation framework. By employing Buffered Flow Matching, it realizes a sliding-window mechanism over diffusion-model denoising timesteps, and further introduces several practical techniques such as MicroStep and WindowAttention. The approach supports interactive streaming generation and is applicable to long video synthesis.

**Strengths:**

The experimental results achieve state-of-the-art performance among the compared methods, and the demo videos in the supplementary material exhibit excellent visual quality.

**Weaknesses:**

There are weaknesses in both the novelty of the proposed method and the completeness of the experiments:
- The proposed method shows limited distinction from prior works such as FIFO-Diffusion [1] and Diffusion Forcing [2], lacking clear methodological novelty. The proposed Buffered Flow Matching shares a highly similar underlying idea with FIFO-Diffusion, differing mainly in that the former is trainable, while FIFO-Diffusion is training-free. In addition, the MixedTraining strategy within each chunk closely resembles the approach used in Diffusion Forcing.
- In terms of experimental evaluation, the paper omits several recent streaming video generation methods, such as CauseVid [3] and Self Forcing [4]. It only compares the proposed approach with ReuseDiffuse and FIFO-Diffusion, where the former is an older 2023 work, and the latter is a training-free method. Given that the proposed model requires training, outperforming these baselines is trivial. If this work could demonstrate that its performance surpasses CauseVid and Self Forcing, it would improve my overall impression of the paper.

A minor concern (not a major factor in the rating) is that this work is built upon a closed-source model and does not provide any code in the supplementary materials. This reduces reproducibility and diminishes the community contribution of the paper.

[1] FIFO-Diffusion: Generating Infinite Videos from Text without Training

[2] Diffusion Forcing: Next-token Prediction Meets Full-Sequence Diffusion

[3] From Slow Bidirectional to Fast Autoregressive Video Diffusion Models

[4] Self Forcing: Bridging the Train-Test Gap in Autoregressive Video Diffusion

**Questions:**

See the Weaknesses section.

---

### Official Review · Reviewer_CuD9 · 2025-11-01

**Soundness:** 3
**Presentation:** 3
**Contribution:** 2
**Rating:** 4
**Confidence:** 4

**Summary:**

This paper introduces StreamDiT, a pipeline for real-time, streaming text-to-video generation. The method builds on a flow matching framework by incorporating a moving buffer with a generalized partitioning scheme. The system combines several components, including chunk partitioning, window attention, and a tailored multistep distillation process to achieve efficient, autoregressive video generation at real-time speeds.

**Strengths:**

- The system appears to be well-engineered and thoroughly implemented.

- The paper conducts extensive ablation studies to demonstrate the effect of each component in the proposed pipeline.

**Weaknesses:**

- The paper's core technical contribution seems somewhat limited. It builds upon the central idea of FIFO-Diffusion (adding varying levels of noise to tokens in a time sequence for autoregressive generation) and introduces modifications like chunk partitioning, window attention, and distillation to create a pipeline for real-time video generation. While this represents a strong engineering effort, the primary techniques used (e.g., window attention, model distillation) are relatively standard for accelerating video generation models. This limits the overall technical novelty.

- The comparison is restricted to two closely related prior works (ReuseDiffuse and FIFO-Diffusion) that also condition trunk diffusion generation on previous frames. Other lines of research, such as (chunk-wise) causal autoregressive models (e.g., Self-Forcing, CausVid)—which also often involve distillation for fast streaming—are not included. While comparing models with different bases can be difficult, the paper would be strengthened by adding a discussion and comparison against these works. This is especially important given that the proposed method is built on a base model that is not publicly available, making it difficult to assess the method's performance and standing within the broader field.

- The proposed pipeline conditions the generation of new frames only on the current chunk. This design fails to capture long-range semantic coherence, which is crucial for long video generation.

**Questions:**

- Related to the Weaknesses, the autoregressive paradigm for long video generation typically suffers from error accumulation and content drift over time. How does the proposed method perform from this perspective?

---

### Note · Authors · 2025-11-14

I have read and agree with the venue's withdrawal policy on behalf of myself and my co-authors.